# Exploring the Relationship between Canine Paraoxonase-1 (PON-1) Serum Activity and Liver Disease Classified by Clinico-Pathological Evaluation

**DOI:** 10.3390/ani14192886

**Published:** 2024-10-08

**Authors:** Sara Meazzi, Sabiha Zarin Tasnim Bristi, Virginia Bianchini, Paola Scarpa, Alessia Giordano

**Affiliations:** 1Department of Veterinary Medicine and Animal Sciences, University of Milan, Via dell’Università 6, 26900 Lodi, Italy; 2I-VET Laboratory, Via E. Majorana 10, 25020 Flero, Italy

**Keywords:** acute-phase protein, canine, liver disease, paraoxonase-1

## Abstract

**Simple Summary:**

This study measures the activity of an enzyme called Paraoxonase-1 (PON-1) in dogs. This enzyme is produced by the liver and may decrease its activity during systemic inflammation. The goal was to observe how PON-1 levels change in dogs with suspected liver diseases. One hundred sixty dogs were enrolled and divided into three groups based on their health status: healthy controls (20 dogs), suspected liver injury (114 dogs), and suspected liver failure (26 dogs). The PON-1 levels in their blood were measured using a specific method. There was no difference in PON-1 levels between the control and suspected liver injury groups. However, the dogs with high serum levels (up to nine times the upper reference limit) of a liver enzyme called alanine aminotransferase, which increases after liver damage, also had higher PON-1 levels. The dogs with jaundice had lower PON-1 levels, while those with fatty blood had higher levels. The dogs in the suspected liver failure group had significantly lower PON-1 levels compared to those in the other groups. The results show that we should be careful when interpreting PON-1 levels in dogs with possible liver issues.

**Abstract:**

Paraoxonase-1 (PON-1), a liver-synthesized enzyme, acts as a negative acute-phase reactant during systemic inflammation in dogs. Given the hepatic synthesis of this enzyme, the presence of liver diseases may influence PON-1, thus affecting its reliability as a biomarker for inflammatory/oxidative systemic diseases. The aim of this study is to investigate PON-1 activity variations among dogs suspected of liver injury or failure, evaluating the influence of hepatic diseases on PON-1 activity. A total of one-hundred-sixty dogs were retrospectively enrolled and categorized into three groups based on clinical presentation and laboratory results: control (C = 20), suspected liver injury (INJ = 114), and suspected liver failure (FAIL = 26). The INJ group was further divided into subgroups based on the severity of the alanine aminotransferase (ALT) increase. Both the INJ and FAIL groups were further divided based on serum macroscopic appearance. The PON-1 activity was quantified using a paraoxon-based method, which is already validated in dogs. No significant difference in PON-1 activity was observed between the C and INJ groups, despite a significant increase in the subgroups with moderate and severe elevations of ALT. The dogs with icteric serum exhibited decreased PON-1 activity, while lipemic serum was associated with an increased PON-1 activity. A significant reduction in PON-1 activity was noted in the FAIL group, compared to both C and INJ groups (*p* < 0.0001), regardless of serum appearance. Given the retrospective nature of this study, additional evaluations (e.g., histopathology, imaging) were not performed. The results obtained here suggest the importance of interpreting PON-1 activity cautiously in dogs with suspected liver disease.

## 1. Introduction

Paraoxonase-1 (PON-1) is a member of the enzyme family possessing paraoxonase, arylesterase, and lactonase activities, being primarily synthesized in the liver [1,2]. In the peripheral blood, PON-1 mainly binds to high-density lipoprotein (HDL) molecules via apolipoprotein-A. A smaller fraction interacts with very low-density lipoproteins (VLDLs), chylomicrons, or circulates freely [3]. Its pivotal function lies in preventing the oxidation of low-density lipoproteins (LDLs) and HDLs by impeding lipoperoxide formation [4]. Additionally, PON-1 can hydrolyze hydrogen peroxide, a highly reactive compound generated during oxidative stress [5].

PON-1 plays a role in the acute inflammatory response as a negative acute-phase reactant in both humans and animals [6,7,8,9,10,11]. In the canine species, PON-1 serves as an inflammatory biomarker, particularly in diseases associated with increased oxidative stress [12], such as babesiosis [13], leishmaniasis [8,14], parvoviral infection [15], dirofilariasis [16], pancreatitis [17], mitral valve disease [18], and lymphoma [19].

Liver disease in the canine species stems from various causes, including infectious, metabolic, toxic, neoplastic and immune causes [20,21]. These conditions may manifest as liver injury, characterized by hepatocyte necrosis, which may occur without impairment of the liver function, or they can induce liver failure, marked by the loss of more than 70% of functional hepatocytes and rapid decline in hepatic function [20,21,22]. Clinical signs of both conditions are non-specific, including anorexia, lethargy, vomiting, diarrhea, jaundice, ascites, and potential neurological signs associated with hepatic encephalopathy. Specific diagnostic procedures, such as imaging, histopathology, or laboratory tests, are necessary for identifying liver injury and failure [22,23]. For liver injury, the elevation in serum activity of the cytosolic hepatocytic enzymes, particularly alanine aminotransferase (ALT), is observed, identifying ALT as the most specific liver injury enzyme in dogs. ALT serum activity increases 12 h post hepatocyte damage and peaks within 24–48 h [23]. An increase in ALT of less than 3 times the laboratory upper reference limit should be interpreted with caution, since it is reported to be more consistent with non-specific reactive hepatopathy rather than hepatocytic inflammation or damage [21]. Conversely, liver failure is characterized by alterations in the metabolites produced, metabolized, or excreted by the liver, such as albumin, urea, total bilirubin, ammonium, bile acids, and coagulation factors [23]. While changes in a single metabolite lack specificity for liver failure, a panel of analytes combined with a consistent clinical presentation strongly indicates its presence. Typical clinicopathological alterations associated with liver failure in dogs include decreased albumin, urea, glucose, and coagulation factors, alongside elevated total bilirubin, ammonium, and bile acids [24].

Considering that PON-1 is predominantly synthesized in the liver [6], it is reasonable to hypothesize that the presence of liver disease could impact the serum activity of this enzyme. Therefore, the aim of this retrospective study is to examine whether a suspected liver disease, as indicated by clinicopathological changes, affects serum PON-1 activity in dogs, thus possibly influencing its accuracy as diagnostic marker for other systemic inflammatory conditions and/or oxidative damage.

## 2. Materials and Methods

All the samples were obtained from diagnostic samples leftover from the Clinical Pathology Laboratory of the Veterinary Teaching Hospital (University of Milan) and stored frozen at −20 °C. The samples were selected from the laboratory database as follows:A control group of healthy dogs, without any clinical signs or clinicopathological abnormalities suggestive of disease (C group);A group of dogs with serum ALT activity exceeding at least twice the laboratory upper reference limit (URL = 60 U/L, ALT > 120 U/L), suggestive of liver injury (INJ group). This group was further categorized into subgroups based on the severity of ALT increase: mild (ALT = 2–3 × URL), moderate (ALT = 4–5 × URL), severe (ALT = 6–9 × URL), or very severe damage (ALT > 10 × URL). Additionally, the INJ group was divided based on the serum macroscopic appearance into normal, icteric, lipemic or hemolytic;A group of dogs with increased serum bile acids (BAs) above the laboratory upper reference limit (URL = 15 µmol/L) and a concurrent decrease in serum albumin (ALB) and urea below the laboratory lower reference limits (LRLs), which is indicative of possible liver failure (FAIL group). Laboratory reference intervals (RIs) were 25–35 g/L for albumin and 3.33–9.99 mmol/L for urea. The group was further divided according to the serum macroscopic appearance into normal, icteric, lipemic, or hemolytic.

For each sample, a serum aliquot of at least 150 μL was available. Since all the samples were leftovers from routine diagnostic purposes, according to the Institutional Ethical Committee, an additional formal approval from the owner was not required (decision EC 29 October 2012, renewed with the protocol n° 02-2016).

All the biochemical analytes were previously measured using the automated spectrophotometer BT3500 (Biotecnica Instruments, Rome, Italy) and the Futurlab Srl reagents and quality control materials (Limena, PD, Italy). Specifically, ALT activity was determined by the IFCC kinetic method without pyridoxal phosphate, urea concentration was measured by the urease method, albumin concentration was measured by the bromocresol green method, bile acids were measured using a colorimetric enzymatic method (3α-hydroxysteroid dehydrogenase). Information about macroscopic serum appearance for each sample was recorded. For macroscopic icteric samples, bilirubin was always measured to confirm the macroscopic data. Bilirubin was measured using the same automated spectrophotometer described above, using the Jendrassik-Grof method.

Serum PON-1 activity was spectrophotometrically measured using an automated analyzer (Cobas Mira, Roche Diagnostic, Basel, Switzerland) with a method already validated in dogs, which uses the organophosphate paraoxon as substrate to ensure that only PON1 and no other enzymes are measured. The rate of hydrolysis of paraoxon to p-nitrophenol was measured by monitoring the increase in absorbance at 540 nm using a molar extinction coefficient of 18,050 L/mol/cm^−1^. A PON-1 activity of 1 U/L was defined as 1 µmol of p-nitrophenol formed per minute under the assay conditions [25].

The statistical analyses were performed using Analyse-it software for Microsoft Excel (Leeds, UK, version 4.97), and the significance level was set at *p* < 0.05. Descriptive statistics were performed for each evaluated parameter, in each group of dogs. The comparison between the group of healthy controls and sick animals (INJ and FAIL taken together) was performed using the Wilcoxon—Mann–Whitney test. The comparison among the INJ, FAIL, and C groups was performed using the Kruskal–Wallis test, followed by a post-hoc Mann–Whitney U-test in case of significant results. The same statistical test was used to compare groups stratified by the macroscopic serum appearance or the severity of liver injury. The correlation between PON-1 and ALT activities was assessed using a Spearman’s test.

## 3. Results

### 3.1. Samples

From the retrospective search, the data and samples from 160 dogs were retrieved, and the patients were divided into the C group (*n* = 20), a sick group that included 140 dogs, subdivided into the INJ group (*n* = 114) and the FAIL group (*n* = 26). The C group was composed of 11 males (55%) and nine females (45%), with a median age of 5 years (min–max = 2–13 years). Most of them were mongrels (13/20, 65%). The INJ group was composed of 58 males (51%) and 51 females (45%). In five cases, gender was not reported. The median age was 11 years (min–max = 6 months–18 years). The dogs were of various breeds, with mongrels being the most represented. Other frequently reported breeds were Labrador retrievers, Shih Tzu, and Cocker. Finally, in the FAIL group there were 10 males (38%) and 16 female (62%). The median age was 8 years (min–max = 3 months–12 years). Signalment and breed distribution of all the three groups are reported in Appendix A.

### 3.2. Comparison of PON-1 among Groups

The comparison between healthy controls and the whole group of sick dogs (INJ + FAIL) did not reveal any significant differences (*p* = 0.73). On the other hand, the statistical analysis among C, INJ, and FAIL groups, revealed the presence of significant differences (*p* < 0.0001). Specifically, significantly lower PON-1 activity was recorded in the FAIL group, compared to both the C group (*p* < 0.0001) and the INJ group (*p* < 0.0001). On the contrary, no significant differences were observed in the comparison between the INJ and C groups (*p* = 0.1038) (Table 1, Figure 1). The values of ALT activity, urea, albumin, bile acids, and PON-1 activity in the entire caseload are reported in Appendix A.

### 3.3. Evaluation of PON-1 in Dogs with Suspected Liver Injury

In the INJ group, serum ALT activity ranged from 120 to 3650 U/L (median 217.5 U/L). There was no significant correlation between PON-1 and ALT activities (*p* = 0.82; r = −0.021). The dogs were stratified according to the liver damage severity into mild (*n* = 43), moderate (*n* = 31), severe (*n* = 19), or very severe damage (*n* = 21). In the severe subgroup, the PON-1 activity was significantly higher compared to all the other subgroups, with the exception of the moderate one (the *p* values of the comparison between the severe subgroup and the control, mild, moderate, and very severe subgroups were 0.014, 0.040, 0.373, and 0.0033, respectively). On the other hand, in the moderate subgroup, the PON-1 activity was significantly higher only when compared to the control and the very severe subgroup (*p* = 0.005 and 0.0026, respectively), but not when compared to the mild and severe subgroups (*p* = 0.089 and 0.373, respectively). There was no significant difference between the mild subgroup and the control (*p* = 0.297) or the very severe subgroup (*p* = 0.112), or between the control and the very severe subgroup (*p* = 0.262) (Table 1; Figure 2).

Finally, samples were further divided based on the serum macroscopic appearance in normal (*n* = 53), icteric (*n* = 10), lipemic (*n* = 38), or hemolytic (*n* = 13). The icteric samples always showed significantly lower PON-1 activity compared to the other subgroups (*p* < 0.0001, compared with lipemic and normal samples, and *p* = 0.0064, compared with the hemolytic samples) and to the control group (*p* = 0.0004). The PON-1 activity in the lipemic samples was always significantly higher compared to the controls (*p* < 0.0001) and to the other subgroups (*p* < 0.0001 when compared with the icteric samples, *p* = 0.0042 when compared with the hemolytic samples, and *p* = 0.0021 when compared with the normal subgroup). No differences were observed between the hemolytic and normal subgroups (*p* = 0.371), nor between the subgroups and the control group (*p* = 0.86 and 0.970, respectively) (Table 1; Figure 3).

### 3.4. Evaluation of PON-1 in Dogs with Suspected Liver Failure

The median values (min–max values) of serum bile acids, urea, and albumin in dogs belonging to the FAIL group were 68.7 µmol/L (15.4–401), 2.66 mmol/L (1.16–3.33), and 20 g/L (13.1–25), respectively. The FAIL group was further divided based on the serum macroscopic appearance in normal (*n* = 16), icteric (*n* = 9), and lipemic (*n* = 1). Given the scarce numerosity of the lipemic subgroup, this was excluded from further statistical analysis. The PON-1 activity of the FAIL group was always lower compared to that of the control group, despite the different serum appearance (*p* < 0.0025 when compared with the normal subgroup, and 0.0001 when compared with the icteric subgroup). The icteric subgroup shows a lower PON-1 activity value compared with the normal subgroup (*p* = 0.0203) (Table 1; Figure 4).

## 4. Discussion

PON-1, which is an enzyme produced by the liver, demonstrates decreased activity in people with liver diseases, especially in patients with chronic disease manifestations, and it has been proposed as a biomarker for evaluating liver function in humans [24]. In veterinary medicine, PON-1 has been used for monitoring infectious diseases, for prognostic purposes [8,16,26,27,28], and, more recently, in non-infectious diseases [17,19,29]. However, information regarding PON-1 activity and liver disease in the canine species is limited in the literature. Given that the liver is the primary source of PON-1 production, diseases affecting the hepatic function may influence PON-1 activity, potentially impacting its accuracy as an oxidative and inflammatory biomarker.

It is interesting to note that when healthy controls were compared to the whole group of dogs with suspected liver injury/failure taken together, no differences in PON-1 activity were observed. The interquartile range of this latter group showed how the data concerning PON-1 activity were widely distributed, suggesting the need for further subclassification based on suspected liver disease. Indeed, the comparison among the three different groups revealed the presence of significantly lower PON-1 activity in the FAIL group compared to the others. The highest PON-1 values were instead recorded in the INJ group. This difference in PON-1 activity could explain the absence of significant results in our first analysis and underlines the need of evaluating PON-1 activity considering these two conditions separately.

In people with hepatopathies, a significant negative correlation between ALT and PON-1 activities exists, with the PON-1 activity decreasing with increased severity of liver damage [24,30,31]. Both PON-1 and ALT are cytoplasmic enzymes; thus, the concurrent increased serum activities of these enzymes could result from hepatocyte leakage during hepatocellular injury [23]. However, hepatocellular necrosis triggers an inflammatory response, along with an increase in free oxygen radicals. These molecules, when interacting with PON-1, lead to its inactivation [32], resulting in a decrease in PON-1 activity during an inflammatory response [33], possibly blunting the effect of the hepatocellular leakage.

In contrast to findings in humans, the results obtained in the present study do not support a correlation between ALT and PON-1 activities in dogs. Furthermore, no significant differences in PON-1 activity were observed in the group of dogs with suspected liver injury compared to the controls. This may be explained by the relatively longer half-life of PON-1, which might delay the decrease in its serum activity, compared to other clinicopathological abnormalities [34]. It is possible that other factors, such as the presence of different polymorphisms that are not investigated in the canine species, influenced the results here obtained. Another possible explanation for this result may rely on the severity of the hepatocellular damage.

Indeed, stratifying dogs into subgroups based on the severity of hepatic damage revealed some significant differences. Specifically, PON-1 activity tends to increase in dogs with liver damage, with significantly higher PON-1 values observed when ALT reaches at least four times the laboratory upper reference limit (URL). This finding may be explained by a PON-1 leakage from damaged hepatocytes. However, in the group with a very severe liver damage (ALT activity at least ten times the laboratory URL), the median PON-1 activity values decreased, showing no significant differences compared to the controls. It is interesting to note that the dogs belonging to this subgroup also showed an increase in bile acids above the laboratory URL. Despite the absence of decreased albumin and urea, this finding may suggest an initial stage of liver failure, possibly associated with extensive liver damage. In those cases, the presence of inflammation associated with liver injury may have an impact on PON-1 activity, possibly counterbalancing the hepatocellular leakage.

In this study, the dogs with suspected liver injury and icteric serum revealed the lowest PON-1 activity values. These animals likely suffered from severe hepatic injury, possibly associated with intrahepatic cholestasis or early liver failure, especially considering that the majority of the icteric samples were from dogs in the very severe subgroup. This result agrees with the literature reporting decreased PON-1 activity in people with both acute and chronic liver injury due to decreased synthesis and strong oxidative stress [31,33]. Analytical interference by bilirubin on PON-1 measurement is unlikely, as no such interference was reported in the method validation study [25]. Therefore, the results here obtained in icteric samples are likely associated with early hepatic failure rather than colorimetric interference. However, due to the limited number of the icteric samples included in this study, further analysis is necessary in a larger cohort of animals to confirm these findings. Unexpectedly, the highest PON-1 activity was obtained in the lipemic samples, as opposed to a false decrease associated with severe lipemia reported in the method validation study [25]. One possible explanation for these results is that in the validation study, lipemia was artificially induced by Lipofundin S 20%, which does not contain cholesterol, and potentially had a different lipid composition than the sera from the dogs with lipemia in the present study. PON-1 predominantly circulates bound to HDLs in serum, acting as a transport protein, while enhancing PON-1 activity [35]. Although it would have been interesting to evaluate the cholesterol and triglycerides concentrations in the serum from the dogs with liver damage, these data were not available for all the samples analyzed due to the retrospective nature of this study.

The dogs with a suspicion of liver failure based on laboratory findings had lower PON-1 activity compared to both the controls and the dogs with suspected liver injury. The concurrent decrease in serum albumin and urea concentrations and the increase in bile acids concentration is suggestive of hepatic failure, reflecting diminished liver synthesis and conjugations capability. The decreased PON-1 activity in this group was expected, since in the literature, a decrease in PON-1 activity was observed in people and laboratory animals with chronic hepatic disease [25,35,36,37]. In the literature, reduced hepatic synthesis has been proposed as the most likely cause for diminished PON-1 activity [24]. However, other studies suggest an association with the increased oxygen free radical concentration or decreased lecithin/cholesterol acytransferase enzyme (LCAT) activity, which is pivotal for cholesterol ester synthesis and HDL formation [33]. The results in this group remain unaffected by serum icteric macroscopic appearance, further supporting the possibility that the results in the FAIL group are primarily due to decreased liver function rather than colorimetric interference.

A potential limitation of this study lies in its retrospective nature. Additional information about other biochemical analytes (cholesterol, triglycerides, ammonia) could have helped evaluate possible correlations between laboratory changes and PON-1 activity. Moreover, the results from histopathology and imaging would help better define the presence and the magnitude of liver damage. Concerning the FAIL group, the decreased albumin may also be associated with its role as a negative acute-phase protein during inflammation; thus, having more information about inflammatory biomarkers (e.g., white blood cell number or C-reactive protein concentration) might help exclude the presence of concurrent inflammation in dogs belonging to this group, where the lowest PON-1 activity values were recorded. For this reason, a serum decreased urea concentration, as well as an increased bile acid concentration, were added as inclusion criteria for the FAIL group, to strengthen the suspicion of liver failure in this group. Prospective investigations, with a complete follow-up of dogs suspected to have liver failure could strengthen more the results obtained in this study.

## 5. Conclusions

Until now, given its antioxidant activity, PON-1 in dogs has mainly been used as a biomarker to identify systemic inflammatory and oxidative conditions. However, the findings presented herein advocate for a cautious interpretation of PON-1 activity values, especially in the case of hepatopathy and, more significantly, when liver failure is suspected. This caution is particularly relevant in icteric samples, where pathological alteration may introduce interference, complicating the accurate interpretation of PON-1 activity results. This becomes especially critical in conditions where systemic inflammation, with an expected decreased PON-1 activity (e.g., canine leishmaniasis, leptospirosis), may coexist with hepatic disease, possibly causing an increase (liver injury) or a further decrease (liver failure) in PON-1 activity. The potential impact of these confounding factors underscores the need for a nuanced approach in evaluating PON-1 as a biomarker in dogs with hepatic disorders, highlighting the necessity for additional investigations to delineate the specific circumstances under which PON-1 activity may be reliably interpreted in the context of liver pathology. The use of PON-1 as a biomarker for liver failure goes beyond the aim of the present study. Nevertheless, the results of this study suggest that a very low level of PON-1 activity may be associated with liver failure in dogs. This hypothesis must be investigated in a larger cohort of dogs with a well-defined diagnosis of liver disease, obtained through histopathology.

## Figures and Tables

**Figure 1 animals-14-02886-f001:**
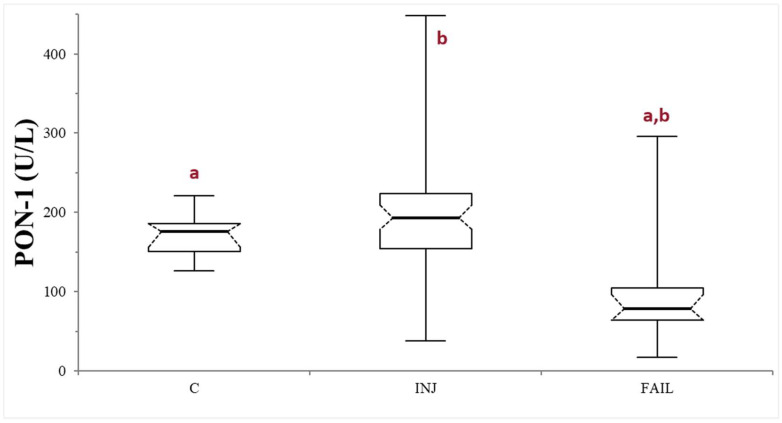
Paraoxonase (PON-1) activity recorded in the control (C), suspected liver injury (INJ), and suspected liver failure (FAIL) groups of dogs. The boxes indicate the first to third interquartile range (IQR), the horizontal lines indicate the median value, whiskers extend to further observation within the first quartile minus 1.5 × IQR or to further observation within the third quartile plus 1.5 × IQR. Significant differences between groups are indicated by the same letter.

**Figure 2 animals-14-02886-f002:**
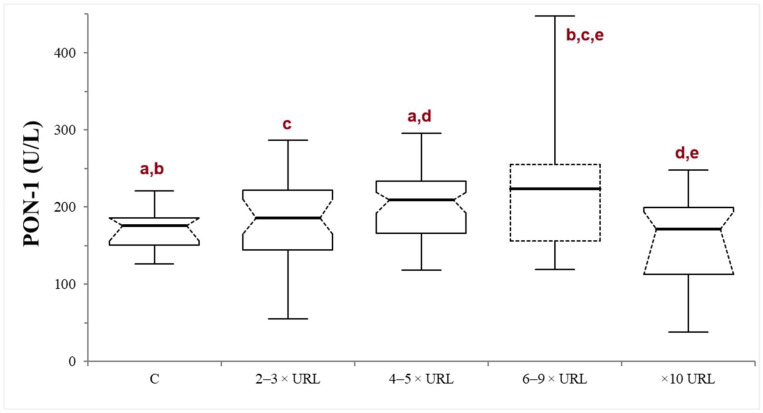
Paraoxonase (PON-1) activity recorded in the control (C) group and in the mild (2–3 × upper reference limit—URL), moderate (4–5 × URL), severe (6–10 × URL) and very severe (>10 × URL) subgroups of dogs with suspected liver injury. The subgroups are divided according to the fold increase of the alanine aminotransferase (ALT) above the laboratory upper reference limit (URL). The boxes indicate the first to third interquartile range (IQR), the horizontal lines indicate the median value, the whiskers extend to further observation within the first quartile minus 1.5 × IQR or to further observation within the third quartile plus 1.5 × IQR. Significant differences between groups are indicated by the same letter.

**Figure 3 animals-14-02886-f003:**
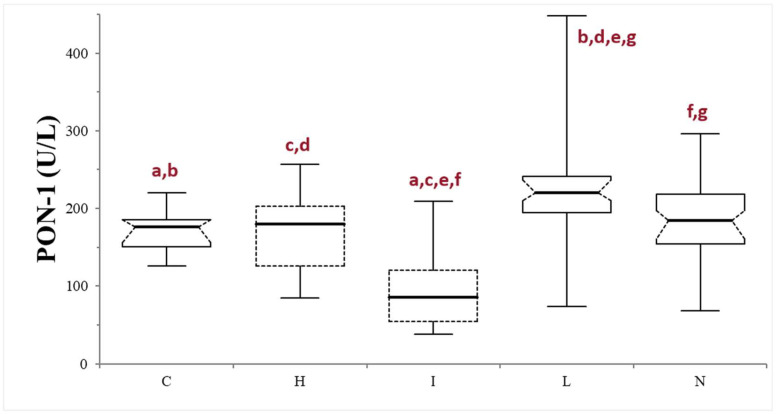
Paraoxonase (PON-1) activity recorded in the control (C) group and in the subgroups of dogs with suspected liver injury divided according to the serum appearance as normal (N), icteric (I), hemolytic (H), and lipemic (L). The boxes indicate the first to third interquartile range (IQR), the horizontal lines indicate the median value, the whiskers extend to further observation within the first quartile minus 1.5 × IQR or to further observation within the third quartile plus 1.5 × IQR. Significant differences between groups are indicated by the same letter.

**Figure 4 animals-14-02886-f004:**
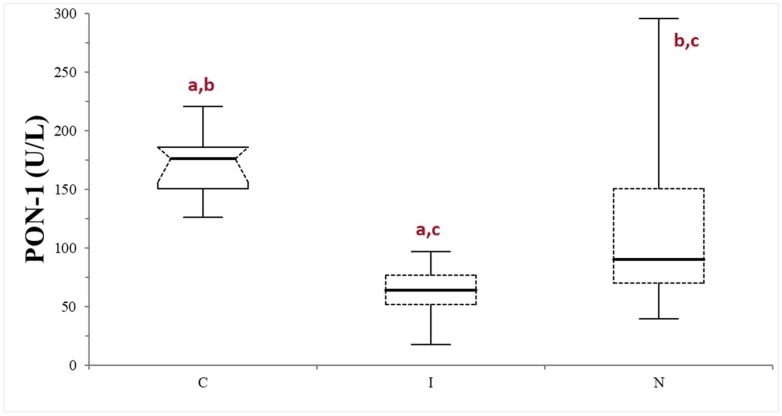
Paraoxonase (PON-1) activity recorded in the control (C) group and in the subgroups of dogs with suspected liver failure, divided according to the serum appearance as icteric (I) and normal (N). The boxes indicate the first to third interquartile range (IQR), the horizontal lines indicate the median value, the whiskers extend to further observation within the first quartile minus 1.5 × IQR or to further observation within the third quartile plus 1.5 × IQR. Significant differences between groups are indicated by the same letter.

**Table 1 animals-14-02886-t001:** Values of Paraoxonase-1 (PON-1) activity, expressed as U/L, recorded in the three groups and in all the subgroups, expressed ad median values, minimum and maximum, and interquartile range (IQR).

Group		Median (Min–Max)	IQR
C (*n* = 20)		176.3 (126.0–220.6)	35.51
INJ + FAIL (*n* = 140)		179.2 (17.5–448.0)	102.1
INJ (*n* = 114)		193.4 (38.3–448.0)	70.02
	ALT 2–3 × URL (*n* = 43)	185.6 (55.7–287.0)	76.94
	ALT 4–5 × URL (*n* = 31)	209.4 (118.1–295.9)	67.73
	ALT 6–9 × URL (*n* = 19)	223.4 (119.6–448)	99.03
	ALT > 10 × URL (*n* = 21)	171.2 (38.3–248.0)	86.37
	Normal (*n* = 53)	184.6 (68.3–295.9)	63.63
	Icteric (*n* = 10)	85.7 (38.3–209.3)	65.6
	Hemolytic (*n* = 13)	180.1 (85–256.6)	76.63
	Lipemic (*n* = 38)	220.15 (73.9–448)	46.44
FAIL (*n* = 26)		78.05 (17.5–296.0)	40.54
	Normal (*n* = 16)	89.85 (39.5–296.0)	80.1
	Icteric (*n* = 9)	63.80 (17.5–96.8)	25.33

IQR = interquartile range; C = control group; INJ = suspected liver injury group; FAIL = suspected liver failure group; URL = upper reference limit.

## Data Availability

The original contributions presented in this study are included in the article/Appendix A; further inquiries can be directed to the corresponding author.

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
