# Peer review of "Exploring the Relationship between Canine Paraoxonase-1 (PON-1) Serum Activity and Liver Disease Classified by Clinico-Pathological Evaluation"

_animals, 2024, doi:10.3390/ani14192886_

Round 1

Reviewer 1 Report (Previous Reviewer 3)

Comments and Suggestions for Authors

The authors have put a great deal of work to explore how liver diseases affect PON1 activity in dogs.

I have some minor considerations (see the comments below).

Introduction

Lines 83-84: I don't understand the aim of the study (especially the second part). Please edit the sentence to make it clearer.

 Results  

All Figures would be easier to follow/understand if you added significance, since you have it.

 Discussion

Lines 228-230: Reference (25) does not appear to belong to the claim made. Please check it out.

Conclusion

Authors reported that PON-1 in dogs  is used as an oxidative biomarker in dogs. Please be specific. PON-1 has antioxidant properties and therefore can be marker of antioxidant capacity, not oxidative. Change it.

Author Response

The authors have put a great deal of work to explore how liver diseases affect PON1 activity in dogs.

I have some minor considerations (see the comments below).

Introduction

Lines 83-84: I don't understand the aim of the study (especially the second part). Please edit the sentence to make it clearer.

AA: the sentence has been changed in agreement with the reviewer’s suggestion, in order to make it clearer

 Results  

All Figures would be easier to follow/understand if you added significance, since you have it.

AA: the significances have been added to the figures and the figure legends have been modified accordingly

 Discussion

Lines 228-230: Reference (25) does not appear to belong to the claim made. Please check it out.

AA: thank for noticing, all the references have been checked out and corrected.

Conclusion

Authors reported that PON-1 in dogs  is used as an oxidative biomarker in dogs. Please be specific. PON-1 has antioxidant properties and therefore can be marker of antioxidant capacity, not oxidative. Change it.

AA. the sentence has been modified based on reviewer’s suggestion

Reviewer 2 Report (New Reviewer)

Comments and Suggestions for Authors

This manuscript has a major limitation which is the lack of a proper characterization of the diseases of the liver. This makes the results of this paper difficult to interprete and generate doubts about its real application in cases of liver failure.

Comments on the Quality of English Language

The english language is appropriate

Author Response

This manuscript has a major limitation which is the lack of a proper characterization of the diseases of the liver. This makes the results of this paper difficult to interprete and generate doubts about its real application in cases of liver failure

AA. We are so sorry to read this since according to previous suggestion, we did a great effort in increasing the likelihood of liver injury and liver failure, reanalyzing samples for the addition of further biochemical parameters, because we totally agree that this would improve the quality of the study. In clinical practice usually veterinarians do not have liver histopathology without previous biochemical analysis. Thus, the aim of this study, clearly stated in the manuscript is to examine whether the presence of a suspected liver disease (indicated by clinicopathological changes) could affect serum PON-1 activity (influencing its accuracy as diagnostic marker for other concurrent systemic inflammatory conditions and/or oxidative damage). We have underlined all the possible limitation referred to the lack of histopathology in the manuscript.

Round 2

Reviewer 2 Report (New Reviewer)

Comments and Suggestions for Authors

It seems that the authors did not solve the limitations raised in the first review

Comments on the Quality of English Language

The english lenguage just requires minor corrections

Author Response

We are so sorry to read this since according to previous suggestion, we did a great effort in increasing the likelihood of liver injury and liver failure, reanalyzing samples for the addition of further biochemical parameters, because we totally agree that this would improve the quality of the study. In clinical practice usually veterinarians do not have liver histopathology without previous biochemical analysis. Thus, the aim of this study, clearly stated in the manuscript is to examine whether the presence of a suspected liver disease (indicated by clinicopathological changes) could affect serum PON-1 activity (influencing its accuracy as diagnostic marker for other concurrent systemic inflammatory conditions and/or oxidative damage). We have underlined all the possible limitation referred to the lack of histopathology in the manuscript.

This manuscript is a resubmission of an earlier submission. The following is a list of the peer review reports and author responses from that submission.

Round 1

Reviewer 1 Report

Comments and Suggestions for Authors

In their manuscript, Impact of liver diseases on paraoxonase-1 (PON-1) serum activity in dogs by Meazzi et al. reported the PON-1 levels in serum (archive samples) of dogs divided within 3 groups (control, suspected liver injury and suspected liver failure). The classification was performed based on alanine aminotransferase and albumin levels, as well as urea concentration. Serum macroscopic appearance was also reported. Suspected injury group contained significantly older animals than another groups. PON-1 activity was correlated to suspected liver disease and serum macroscopic appearance.

The manuscript was clearly written and all the methods described. Statistical analysis was performed entirely satisfactory and the results were clearly presented. 

However, the study design is completely ineffective. The aim of the research was to evaluate the influence of hepatic diseases on PON-1 activity. However, the disease was not confirmed – it is stated SUSPECTED liver disease. The authors performed non-specific laboratory diagnostics to determine the severity of liver disease. More specific tests of liver function were not performed (cholesterol, total bilirubin, or bile acid concentration), as the authors stated the archive samples were used. No imaging and/or histopathology data were provided, which is used as the gold standard for obtaining a definitive diagnosis of liver disease.

The technical confirmation of PON-1 enzyme activity and its correlation to any possible/suspected (instead of exact) diagnosis could not lead to any substantial conclusions about the pathology or biology of the liver disease, no matter how good the experimental data.

The experimental design should include statistically significant number of animals with diagnosed disease in each group as the starting point, and then data correlating the PON-1 activity and the suspected disease would be for example interesting for setting the possible marker (or one of the markers) for the liver disease severity to avoid histopathology (if applicable, regarding the influence of sample serum macroscopic appearance – lipemic or haemolytic samples). For this, lipid profile in serum would be also important, and the authors stated no such data for the samples used herein exist.

For abovementioned reasons, I propose the manuscript to be rejected in its current form. It is clear the authors are aware of all those issues, as explained in Potential limitations – but those are substantial, not potential.

Reviewer 2 Report

Comments and Suggestions for Authors

Line 9: This enzyme is produced by the liver and helps with systemic inflammation.

helps during systemic inflammation? What does it help with? When was the parameter increased? The sentence needs to be reworded because it is incomprehensible and contains mental shortcuts.

Line 14: “High levels of a liver enzyme called alanine aminotransferase had higher PON-1 levels.” There are a couple of liver enzymes. The word was used inadvertently.

Line 52-54: Liver disease in the canine species stems from various causes, including canine adenovirus-1 or Leptospira spp. infection, exposure to toxic compounds like xylitol, or administration of drugs such as acetaminophen [20,21].

fragments of phrases that are not in context. Kindly rank by most recent ACVIM orders. Divide into infectious diseases, non-infectious diseases, etc.

Lines 60–70 Liver diseases are often assessed based on ALT and ASP.

It is worth emphasizing that an increase of several times (at least 4) indicates inflammation of the liver and not irritation, e.g., caused by a poorly balanced diet.

According to ACVIM, over 60% of dogs with chronic hepar disease had increased levels of ALT,ALP,AST, GGT, and TSBA.

Compare these parameters with PON-1.

Division into chronic hepatitis (AH) and acute hepatitis (CH) seems to be more accurate. Why suspected liver injury and suspected liver failure.

Insufficient biochemical information has been available to compare liver disease indicators.

Comments on the Quality of English Language

f. e Line 9: during_> with (however, the entire sentence needs to be corrected for substantive reasons)

dogs with suspected-> without "with" (however, the entire sentence needs to be corrected for substantive reasons)

Reviewer 3 Report

Comments and Suggestions for Authors

The article is interesting and provides valuable information regard impact of liver diseases on PON1 activity in dogs.

I have some minor issues that can be corrected to improve the quality of the manuscript.

The activities of PON1 encompass paraoxonase, lactonase, and arylesterase, all of which can be gauged using diverse substrate. In this study authors used paraoxon-based method to measure serum PON-1 activity.

Is it feasible to obtain different results (eg difference in PON1 activity between C and INJ groups) by utilizing a substrate other than paraoxon? Please bring it up in the discussion.

Results
The results obtained are difficult to follow, please mark in all Figures the significance (where it exists) above column (eg different letters indicate significance).

Discussion
Lines 223-232: I am not satisfied with the explanation. Is the increase in PON-1 activity expected by the authors? Support with a reference or additional explanation (lines 228-232).